# Pathological Study of Demyelination with Cellular Reactions in the Cerebellum of Dogs Infected with Canine Distemper Virus

**DOI:** 10.3390/v16111719

**Published:** 2024-10-31

**Authors:** José Manuel Verdes, Camila Larrañaga, Guillermo Godiño, Belén Varela, Victoria Yozzi, Victoria Iribarnegaray, Luis Delucchi, Kanji Yamasaki

**Affiliations:** 1Pathology Unit, Department of Pathobiology, Faculty of Veterinary, Universidad de la República (Udelar). Route 8 Km 18, Montevideo 13000, Uruguay; camilalarranaga@gmail.com (C.L.); guille030599@gmail.com (G.G.); belenvarela42@gmail.com (B.V.); victoriayozzi@gmail.com (V.Y.); yamasaki-kanji1914@outlook.jp (K.Y.); 2Microbiology Unit, Department of Pathobiology, Faculty of Veterinary, Universidad de la República (Udelar), Montevideo 13000, Uruguay; victoria.iribarnegaray@pedeciba.edu.uy; 3Department of Clinics & Veterinary Hospital, Small Animals Medicine Unit, Clinical Neurology, Faculty of Veterinary, Universidad de la República (Udelar), Montevideo 13000, Uruguay; ldelucchi@fvet.edu.uy; 4Japan International Cooperation Agency (JICA), Br. Artigas 417 Of.601, Montevideo 11300, Uruguay

**Keywords:** canine distemper, demyelination, glial cells, pathology, Purkinje cells

## Abstract

The purpose of this study was to examine the relationship between demyelination and cellular reactions in the cerebellum of Canine Distemper Virus (CDV)-infected dogs. We subdivided the disease staging by adding the degree of demyelination determined by Luxol Fast Blue staining to the previously reported disease staging from the acute stage to the chronic stage, and investigated the relationship between demyelination in the cerebellum and the number and histological changes in astroglia, microglia, and Purkinje cells in each stage. Reactions of astrocytes and microglia were observed at an early stage when demyelination was not evident. Changes progressed with demyelination. Demyelination initially began in the medulla adjoining the fourth ventricle and gradually spread to the entire cerebellum, including the lobes. CDV immune-positive granules were seen from the early stage, and inclusion bodies also appeared at the same time. CDV immune-positive reaction and inclusion bodies were observed in astrocytes, microglia, neurons, ependymal cells, and even leptomeningeal mononuclear cells. On the other hand, infiltration of CDV-immunoreactive particles from the pia mater to the gray matter and further into the white matter through the granular layer was observed from an early stage. Purkinje cells decreased from the intermediate stage, and a decrease in cells in the granular layer was also observed. There was no clear association between age and each stage, and the stages did not progress with age.

## 1. Introduction

Canine distemper is an infectious disease caused by a Morbillivirus, belonging to the Paramyxoviridae family, which infects a broad range of terrestrial and aquatic carnivores [1]. It has been reported that demyelination is a characteristic change in the brains of dogs infected with Canine Distemper Virus (CDV), and it has been discussed that glial cells such as astrocytes, oligodendroglia, and microglia are related with demyelination [1,2,3,4,5]. Many studies have focused on changes in astrocytes during the early stages of CDV infection [6,7,8], and Kabakci et al. [9] reported that CDV infected both astrocytes and oligodendrocytes, and the gradual loss of oligodendrocytes was most likely responsible for the progressive demyelination. In addition, demyelination during CDV-demyelinating lesions was said to represent a biphasic process with a primary virus-induced oligodendroglial dystrophy followed by a secondary wave of immune-mediated myelin destruction [10,11]. Two possible routes for demyelination have been reported; one route is that CDV infects astrocytes, which then infect oligodendroglia, and the other route is that CDV infects microglia through astrocytes and ultimately affects oligodendroglia [3]. It has been well known that CDV primarily affects the cerebellum [5,12,13]. On the other hand, studies using gene expression analysis of astrocytes after infection have also been reported recently, as astrocytes are the main cellular target of CDV and already undergo reactive changes in brain lesions before demyelination [2].

Histopathologically, cerebellar lesions have been classified based on Hematoxylin-Eosin (HE)-stained specimens, with the addition of immunohistochemistry, transcriptome, and various other methods [3,8,10,14,15,16]. On the other hand, it has been reported that CDV infection experiments on ferrets preferentially target neurons in general, and particularly Purkinje cells and granule cells [17]. The purpose of this research is to clarify the relationship between demyelination and glial cells, using the cerebellum of natural CDV-infected dogs, in which the presence of the virus was confirmed by immunohistochemistry against CDV, and positive Polimerase Chain Reaction (PCR) tests. Furthermore, we subdivided the stage classification by adding the degree of demyelination using Luxol Fast Blue (LFB) staining to the conventional classification from acute-to-chronic stage, and examined the relationship between demyelination and each glial cell in each stage. Finally, we investigated the Purkinje cell changes in each of these stages.

## 2. Materials and Methods

### 2.1. Dogs

Twenty-eight dogs naturally infected by CDV were used in this study. The ages ranged from 25 days to 10 years. There were 15 male and 13 female dogs, respectively. CDV-immunopositive granules were confirmed in several regions of the brains of the dead dogs through immunohistochemical testing, and infection in twenty-eight dogs was confirmed through PCR analysis. In addition, considering the age of CDV-infected dogs, five dogs ranging from 45 days old to 6 years old were selected as a control group. In the control group, infectious diseases were excluded as the cause of death, and immunohistochemical or PCR tests were negative.

In our previous study on natural CDV-infected dogs [18], we classified the stages of the histopathological lesions as acute, subacute, or chronic based on previous studies on natural and experimental infection in dogs [3,11,14,15,16,19,20]. Namely, acute lesions were characterized by focal vacuolation, gliosis, inclusion bodies, and CDV-immunopositive cells without demyelination, subacute lesions were characterized by demyelination, gliosis, necrosis, inclusion bodies, CDV-immunopositive cells, and perivascular mononuclear infiltration (cuffing) of two to three layers of thickness, and chronic lesions were similar to the subacute stage, but perivascular infiltration was more prominent, of at least three layers of thickness. In the present study, we subdivided subacute changes based on the degree of demyelination determined by LFB staining in order to clarify the relationship between demyelination and other changes. In the acute group, no demyelination was observed, in subacute 1 group, demyelination was less than 30% of the entire specimen, in subacute 2 group, demyelination was in 30 to 70%, and in subacute 3, demyelination was 70% or more. We classified the disease into acute, subacute 1, subacute 2, subacute 3, and chronic groups based on HE staining and LFB staining. As a result, the 28 dogs of CDV infection were subdivided into 4 acute cases, 4 subacute 1 cases, 9 subacute 2 cases, 5 subacute 3 cases, and 6 chronic cases.

### 2.2. Histopathology and Immunohistochemistry

Routine autopsy was performed soon after death and selected tissues including the cerebellum were fixed in 10% buffered neutral formalin solution and routinely processed for histologic examination. Tissue sections were stained with Hematoxylin-Eosin (HE), as well as with LFB stains according to Verdes et al. [21].

For immunohistochemistry, a mouse anti-CDV monoclonal antibody (Biorad, MCA 1893, Hercules, CA, USA) was used, followed by a conjugated secondary antibody in an HRP-polymer detection system (Mouse-on-canine HRP-polymer, Biocare Medical, Pacheco, CA, USA); positive antigen–antibody reactions were observed by incubation with 3.3-diaminobenzidine-tetrahidrochloride (DAB) as described previously [22]. Glia cells were immunostained using specific primary antibodies against intermediate filaments of astrocytes (GFAP) and microglia (Iba1). According to Verdes et al. [23], a primary antibody against the calcium binding protein Calbindin D 28k (CbD28k) was used as a marker to identify and count Purkinje cells. Details of the type and origin of primary antibodies, dilution used, and sources are summarized in Table 1.

In addition, four pathologists (JMV, CL, BV, KY) classified the stages of the histopathological lesions following as acute, subacute, or chronic based on the previous studies [3,10,11,14,15,16,19,24]. Acute lesions were characterized by focal vacuolation, gliosis, inclusion bodies, and CDV-positive cells, subacute lesions were characterized by demyelination, gliosis, necrosis, perivascular mononuclear infiltration (cuffing) of two to three layers of thickness, inclusion bodies, and CDV-immunopositive cells, and chronic lesions were similar to the subacute stage, but perivascular infiltration was more prominent, of at least three layers of thickness.

Changes in the numbers of astrocytes, microglia, and Purkinje cells were examined based on the results of the above immunohistochemical staining. The area to be examined was the white matter adjacent to the fourth ventricle for astroglia and microglia, where the changes first appeared and where the changes were characteristic. Purkinje cells were evaluated in the cerebellar cortex, selecting the neighbor cerebellar folia to the same anatomical region studied to evaluate glial cells.

### 2.3. Image Capture and Analysis

All histological slides were scanned with a slide scanner (Motic Easy Scan One^®^, Motic China Group Co., Ltd., Xiamen, China) for subsequent analysis, and the whole slide images were viewed and captured using the Motic DSAssistant^®^ software (Motic VM V1 Viewer 2.0^®^, version 2019-08-02, China).

Astrocytes, microglia, and Purkinje cells were counted in canine cerebellar sections, based on the results of the specific immunohistochemistry against GFAP (astrocytes marker), Iba1 (microglia marker), and CbD28k (Purkinje cell marker).

The counting was done manually in images captured in 400× fields for astrocytes and microglia. The examined region included the fourth ventricle region and the central white matter, capturing two images per animal, one for each area. For the counting of Purkinje cells, a whole image of a cerebellar folia per animal was used, in fields viewed at 40×.

### 2.4. Statistical Analysis

Statistical analysis was performed using GraphPad Prism^®^ (version 10.1.1 -323- for Windows 64-bit, GraphPad Software, Boston, MA, USA, www.graphpad.com). One-way ANOVA tests followed by multiple comparisons using Tukey’s test were used to assess the difference between groups. Data are presented as means ± standard deviation. The level of statistical significance used in all studies was *p* ≤ 0.05.

## 3. Results

### 3.1. Relationship Between Each Stage and Age or Sex

There was no relationship between the stage and the dog’s age, and there was no tendency for the acute to be young and the chronic to be old. For example, an animal was died at 45 days of age in the chronic stage. There was also no relationship between each stage and sex. Many of the dogs were mixed breed, and there was no relationship between each stage and a specific breed. In the control group, the numbers of astrocytes, microglia, and Purkinje cells were not related to age, breed, sex, or body weight.

### 3.2. Histopathological and Immunohistochemical Findings

A summary of histological findings is shown in Table 2.

A summary of the immunohistochemical study is shown in Table 3; the graphical representation of the total number of GFAP-immunoreactive astrocytes in each stage, with statistically significant differences observed between the control and chronic group of *p* = 0.0002 (Figure 1a), and the representative immunohistochemistry against GFAP in the cerebellum of a normal dog (Figure 1b) and the cerebellum of a dog in the chronic stage (Figure 1c) is shown. In Figure 2, the graphical representation of the total number of Iba1-immunoreactive microglia in each stage, without statistically significant differences between the groups (Figure 2a), and the representative immunohistochemistry against Iba1 in the cerebellum of a dog of subacute group 2 (Figure 1b,c) are shown. Figure 3a shows the graphical representation of the total number of CbD28k-immunoreactive Purkinje cells in each stage, with statistically significant differences observed between the control and subacute group 2 (*p* = 0.0144), control and subacute 3 (*p* = 0.0024), and between the control and chronic groups (*p* = 0.0163). The representative immunohistochemistry against CbD28k-immunoreactive Purkinje cells in the cerebellum of a normal dog (Figure 3b) and the cerebellum of a CDV-infected dog from subacute 3 stage (Figure 3c) is shown.

LFB staining at each stage is shown in Figure 4a–f, demyelinating changes in white matter are shown in Figure 5, and CDV-immunoreactive particle deposition is shown in Figure 6a–d.

Histologically, the main lesions were demyelination, glial cell reaction, reduced number of Purkinje cells, and meningitis.

In the acute group, all animals subjected to PCR testing were positive. No demyelination and vacuoles were observed by HE and LFB staining. CDV particles were detected in astrocytes, microglia, neurons, pia mater, and the choroid plexus. In addition, CDV-immunoreactive particles were observed in the white matter adjacent to the fourth ventricle in all cases, and were also observed in endothelial cells of blood vessels, mononuclear cells inside and outside blood vessels in the white matter away from the fourth ventricle, the white matter of cerebellar lobes, and the granular layer. Nuclear and cytoplasmic inclusions were observed mainly in astrocytes, microglia, and pia mater. Although there was no significant increase in the number of astrocytes and microglia compared to those in the control group, they were enlarged compared to cells in the control group. CDV-immunoreactive granules were observed in these enlarged cells.

In subacute 1 group, changes were similar to those in the acute stage, but demyelination was apparent by LFB staining. Changes were occurring in the white matter adjacent to the fourth ventricle. In one case, CDV-immunoreactive granules infiltrated from the pia mater into the gray matter, and were also observed in Purkinje cells, and in cells of the granular layer. Although there was no significant increase in the number of astrocytes and microglia compared to the acute group, many enlarged cells were observed.

In subacute 2 group, demyelination occurred frequently in areas other than the white matter adjacent to the fourth ventricle, and demyelination in the white matter of the cerebellar folia was observed in more than half of the cases. Demyelination and associated changes in astrocytes and microglia progressed around blood vessels, such that areas of demyelination were scattered throughout the white matter. Although there was no significant increase in astrocytes and microglia compared to the acute and subacute 1 groups, gemistocytic astrocytes, multinuclear astrocytes, fibrillary gliosis, and rod cells were observed. Foamy cells also appeared in areas where changes were progressing. LFB staining revealed swelling, collapse, and disappearance of the myelin sheath around the myelinated axons. CDV-immunoreactive particles and inclusion bodies were increased compared to the subacute 1 group.

In the subacute 3 group, the changes in the subacute 2 group progressed, and the area of demyelination increased and spread throughout the white matter. Although no significant increase in the number of astrocytes was observed, many gemistocytic astrocytes and multinuclear astrocytes were observed (Figure 7). In addition, CDV particles in inclusion bodies were increased. Although lymphocytes, plasma cells, monocytes, and macrophages were aggregated around blood vessels, the number of these cells was 1 to 2 layers.

In the chronic group, changes such as severe demyelination, a significant increase in the number and enlargement of astrocytes and microglia, an increase in CDV-immunoreactive particles, and an increase in inclusion bodies were progressing. The perivascular cuffing showed three or more layers.

Meningitis was observed in one subacute 1, three subacute 2, three subacute 3, and three chronic cases. Mononuclear cells with CDV-immunoreactive particles were detected in and around blood vessels.

As described, astrocytes and microglia did not increase in number compared to the control group, but were enlarged in the acute group. Furthermore, gemistocytic astrocytes, multinuclear astrocytes, fibrillary gliosis, and rod cells were observed in the subacute groups. Demyelination was observed in the white matter of the cerebellar folia depending on the stages, and CDV infiltration from the pia mater into the gray matter was also observed. According to these changes, a decrease in Purkinje cells and necrotic changes in the granular layer were detected (Figure 8).

## 4. Discussion

Our results showed that in the cerebellum, demyelination, reactions of astrocytes, oligodendroglia, and microglia, the appearance of inclusion bodies, a reduction in the number of Purkinje cells, and degeneration of the granular layer were observed. Demyelination was evident with LFB staining, and was observed as a decrease in staining with HE staining in the subacute 1 group. Demyelination was consistent with vacuolization in the subacute 2 group and later. Astrocytes and microglia were hypertrophic even in the acute stage, when no demyelination was observed, and progressed with the stage. It has been reported that in demyelinating distemper lesions, the majority (95%) of infected cells have been identified as astrocytes, representing the main target for CDV [25,26]. In this study, CDV-immunoreactive granules were observed not only in astrocytes but also in microglia at the early stage of infection. Furthermore, it has been said that astrocytes and microglia responded during the early stages of infection, and there are also reports that their numbers increased [2,3,20]. In this study, there was no significant increase in the number of astrocytes and microglia between the control group and each infection group, but these cells changed morphologically from an early stage, and CDV-immunoreactive granules were also observed in these cells. It is clear that normal function is impaired. The fact that morphological changes, rather than increases in the number of astrocytes and microglia, were a characteristic feature is considered to be important information obtained in this study. Various functions of astrocytes have been reported, among which the key function of astrocytes is said to be the removal of neurotransmitters, such as glutamate, from the synaptic cleft by specific transporters and subsequent degradation by glutamine synthetase (glutamate–glutamine cycle), thus preventing excitotoxic cell death of neurons and myelin-producing oligodendrocytes [16,27]. In this study, astrocyte responses were observed before demyelination, so it is apparent that astrocytes are involved in myelin changes from the early stage of infection.

Cytoplasmic and intranuclear inclusion bodies could be frequently found in acute and subacute lesions [28]. In the present study, CDV granules and inclusion bodies were observed from the acute stage, before the onset of demyelination, and progressed with the stage. It has been reported that viral antigens may disappear from the inflammatory demyelinating lesions due to the antiviral immune responses in chronic CDV encephalomyelitis [29,30]. However, the responses of astrocytes and microglial cells, the increase in the number of CDV-immunoreactive particles within these cells, and the appearance of inclusion bodies did not decrease until the chronic stage in this study. Therefore, it was clear that changes in astrocytes and microglial cells appeared at an early stage and progressed with the stage. Two possible routes for demyelination have been reported; one route is that CDV infects astrocytes, which then infect oligodendroglia, and the other route is that CDV infects microglia through astrocytes and ultimately affects oligodendroglia [3]. Our results do not support either of these possibilities.

Changes were initially observed in the periventricular region and gradually became perivascular in the white matter. The periventricular region is said to be the first area that the virus infects. The main route of neural entry is via infected mononuclear cells that cross the blood–brain barrier, resulting in local viral release and subsequent infection of resident epithelial and endothelial cells. It has also been reported that the virus can spread through the brain once it enters the brain. It infects the cerebrospinal fluid (CSF), infects the ependymal lining cells of the ventricles, and ultimately infects glial cells and neurons [19,31,32,33,34]. On the other hand, demyelination, astrocyte and microglial reactions, and myelin degeneration were also observed in the white matter of cerebellar folia in the subacute 2 group. Furthermore, in the subacute 1 group, infiltration of CDV-immunoreactive granules from the leptomeninges into the gray matter was observed. In these cases, an astrocytic response was also observed in the gray matter, and cell loss in the granular layer was also observed. In cerebellar folia, CDV-immunoreactive particles were found in astrocytes, Golgi cells, and Purkinje cells. CDV-immunoreactive granules have been reported to extend from pial cells to the subpial gray matter [5,13,28,34,35]. We confirmed CDV infection from the leptomeninges to the gray and white matter described in our previous report [18].

In this study, we observed changes to Purkinje cells. The number of Purkinje cells decreased more significantly than in the subacute 2 group and gradually progressed. This fact was thought to be due not only to the influence of CDV infection in the white matter adjacent to the gray matter, but also to CDV infection from the leptomeninges to the gray and white matter. Effects on Purkinje cells have been reported in viral infections in various species of animals [7,36]. Focal vacuolation and a spongy appearance near the Purkinje cell layer in the cerebellum have also been observed in dogs with CDV infection [4], and Rudd et al. [17] also stated that Purkinje cells and granule cells are specifically targeted by CDV infection. In addition, changes in Purkinje cells and granular layers have been reported in CDV infection in other animal species [37], and the changes in Purkinje cells and granule cells have been observed in the CDV infection experimental study in ferrets [17]. Based on the above reports and the present results, changes in Purkinje cells and the granular layer can be said to be one of the characteristics of CDV infection.

Age and each stage of disease have been discussed based on the dog’s immune status. Young or immunocompromised dogs are said to show acute lesions, while mature dogs usually develop the chronic encephalomyelitis [1,11,20,38]. This study did not measure antibodies in the blood. Furthermore, there may be an issue with the strain of the CDV virus. In this study, the samples in the acute group ranged from 25 days after birth to 3 years old, and the onset at age 10 was identified in the subacute 2 group, instead of in the chronic group. These facts suggest that the stage of the lesions is related to the immune status due to various factors, and it seems unlikely that age alone is a factor associated with each stage.

## 5. Conclusions

Astrocyte and microglial responses were observed at an early stage when demyelination was not evident. CDV-immunoreactive granules and inclusion bodies were also observed in various cells from the acute stage. A decrease in Purkinje cells and degenerative changes in the granular layer were also observed with demyelination. These changes were progressing along with the stage. There was no clear relationship between age and stage.

## Figures and Tables

**Figure 1 viruses-16-01719-f001:**
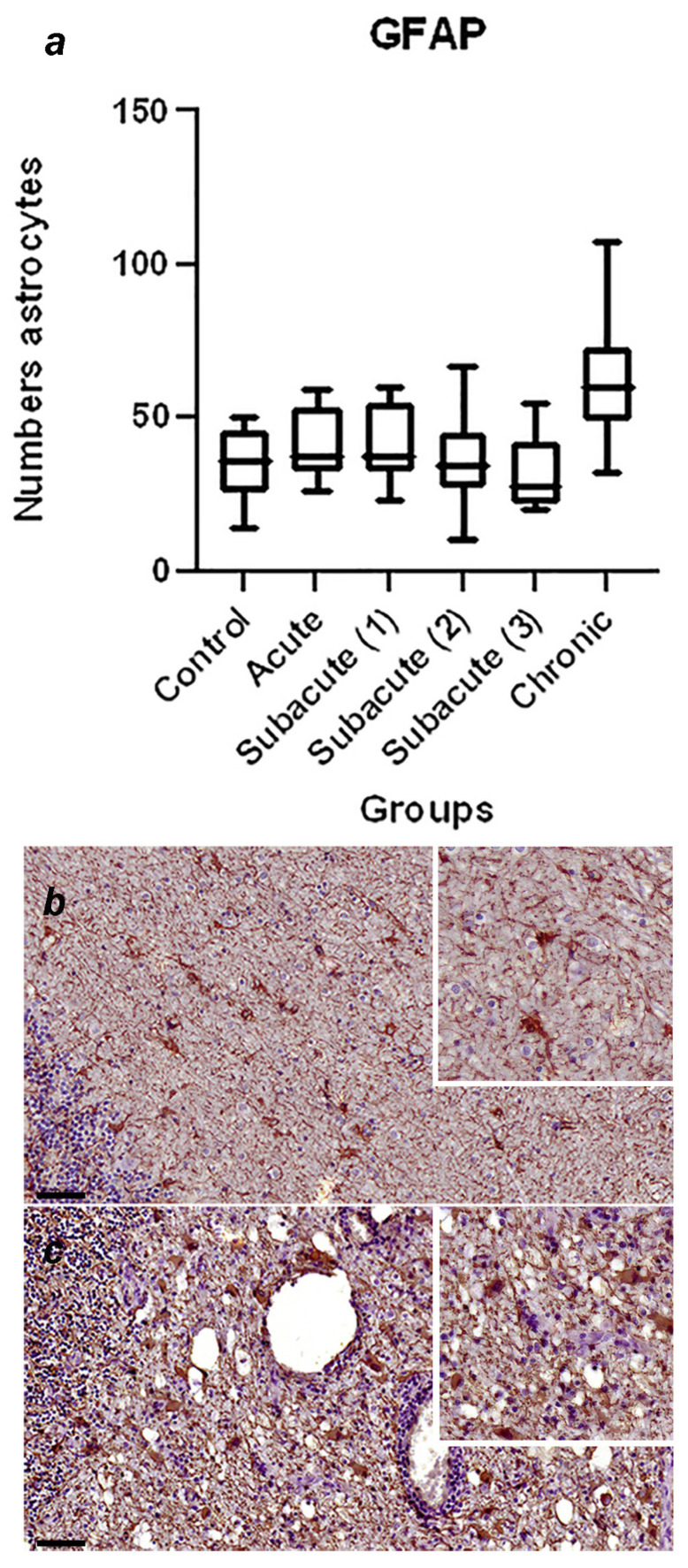
(**a**) Comparative number of astrocytes with positive immunostaining against GFAP in the fourth ventricle and central white matter region between the control, acute, subacute 1, subacute 2, subacute 3, and chronic animal groups. Statistically significant differences were observed between the control and chronic group of *p* = 0.0002. (**b**) Representative GFAP-immunohistochemistry of cerebellar white matter of a normal dog (control group). Immunolabeling for GFAP is observed in normal astrocytes. Scale = 50 μm. See details in the upper right inset of the figure. (**c**) Representative GFAP-immunohistochemistry of cerebellar white matter of a CDV-infected dog (chronic group). Immunostaining for GFAP is observed in reactive astrocytes, showing an increase in cytoplasmic volume in gemistocytic astrocytes. Scale = 50 μm. See details in the upper right inset of the figure.

**Figure 2 viruses-16-01719-f002:**
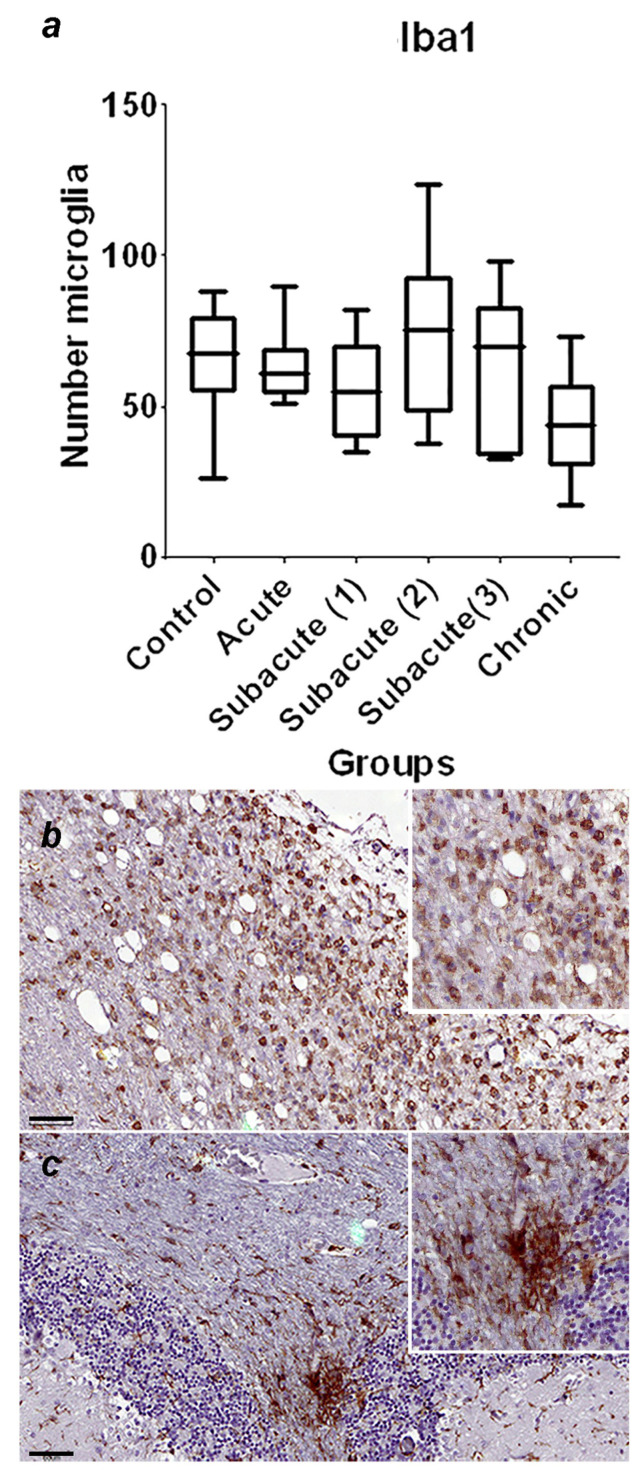
(**a**) Comparative number of microglia with positive immunostaining against Iba1 in the region of the fourth ventricle and the central white matter between the control, acute, subacute 1, subacute 2, subacute 3, and chronic animal groups. No significant differences were found. (**b**) Representative Iba1-immunohistochemistry of cerebellar white matter of a CDV-infected dog from subacute group 2. Immunostaining against Iba1 is observed in the periventricular region, and the microglia have an amoeboid shape. Scale = 60 μm. See details in the upper right inset of the figure. (**c**) Representative Iba1-immunohistochemistry of cerebellar white matter of a CDV-infected dog (chronic group). Same animal and region of (**b**). Intense immunostaining against Iba1 is observed in the cerebellar white matter. Scale = 60 μm. See details in the upper right inset of the figure.

**Figure 3 viruses-16-01719-f003:**
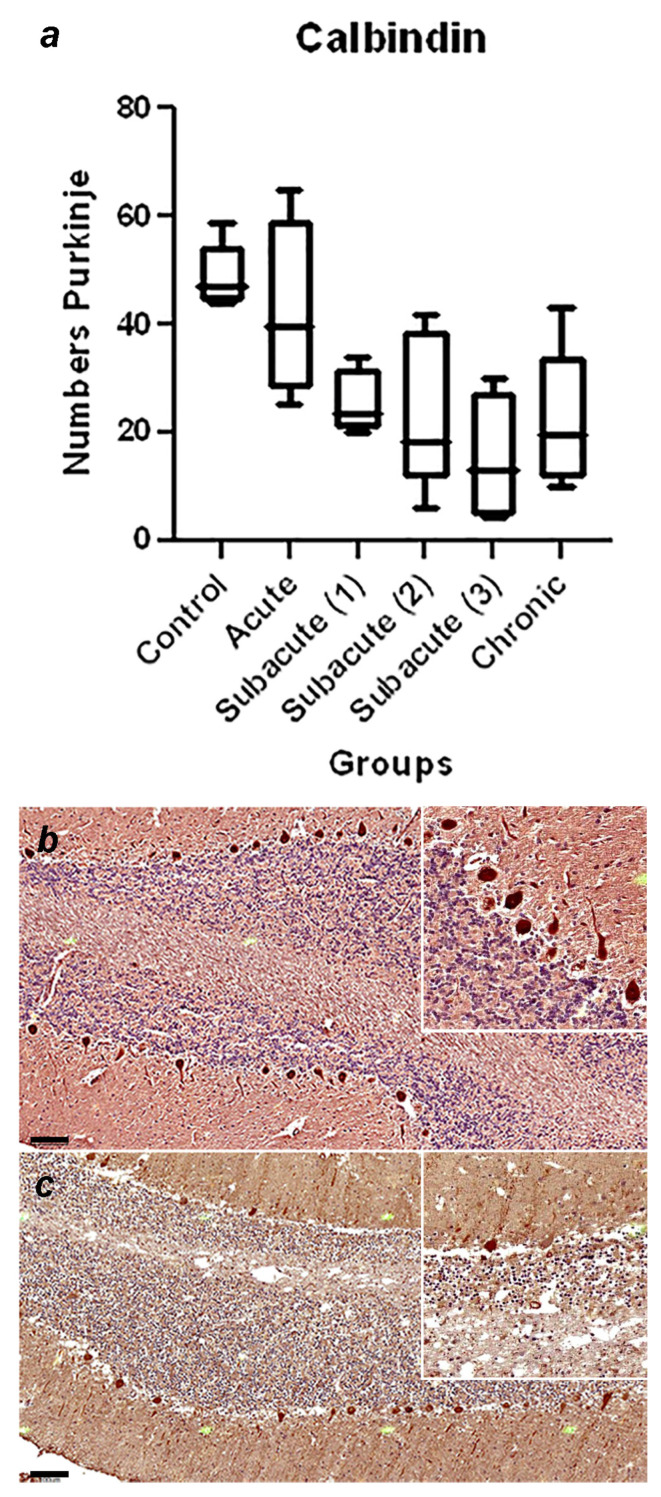
(**a**) Comparative number of Purkinje cells with positive immunostaining against CbD28k in the cerebellar cortex between the control, acute, subacute 1, subacute 2, subacute 3, and chronic animal groups. Statistically significant differences were observed between the control and subacute group 2 (*p* = 0.0144), control and subacute 3 (*p* = 0.0024), and between the control and chronic groups (*p* = 0.0163). (**b**) Representative Calbindin (CbD28k)-immunohistochemistry of cerebellar cortex of a normal dog (control group). Positive immunostaining against CbD28k is observed in all the Purkinje cells (in the soma, axonal process, and dendritic arborization). Scale = 100 μm. See details in the upper right inset of the figure. (**c**) Representative CbD28k-immunohistochemistry of cerebellar cortex of a CDV-infected dog from subacute group 3. Positive immunostaining against CbD28k is observed in all Purkinje cells, showing a reduction in the number of immunoreactive Purkinje cells. Scale = 100 μm. See details in the upper right inset of the figure.

**Figure 4 viruses-16-01719-f004:**
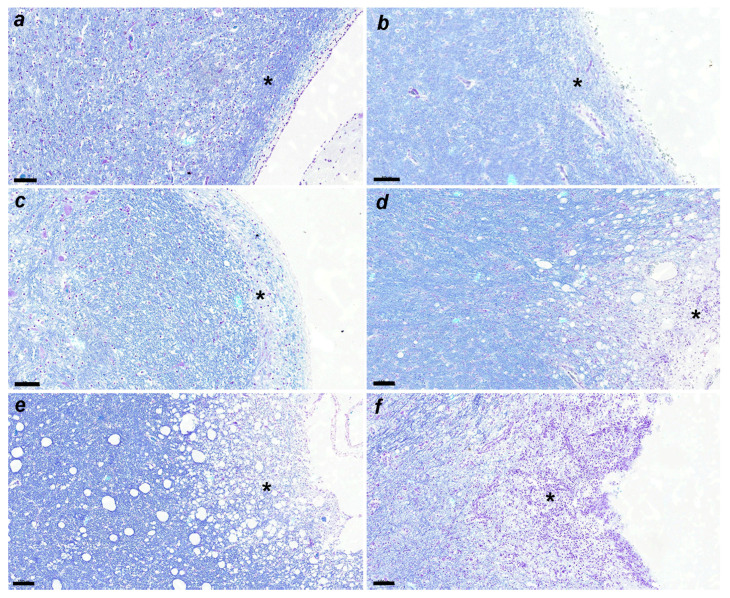
(**a**) Demyelination is not detected. No. 397. Control group. LFB. Scale = 60 μm. See the asterisk. (**b**) Demyelination is not apparent. No. 010. Acute group. LFB. Scale = 60 μm. See the asterisk. (**c**) Slight demyelination is observed. No. 366. Subacute 1 group. LFB. Scale = 60 μm. See the asterisk. (**d**) Demyelination is apparent. No. 463. Subacute 2 group. LFB. Scale = 100 μm. See the asterisk. (**e**) Demyelination is progressed. No. 23. Subacute 3 group. LFB. Scale = 100 μm. See the asterisk. (**f**) Demyelination is severe. No. 11. Chronic group. LFB. Scale = 100 μm. See the asterisk.

**Figure 5 viruses-16-01719-f005:**
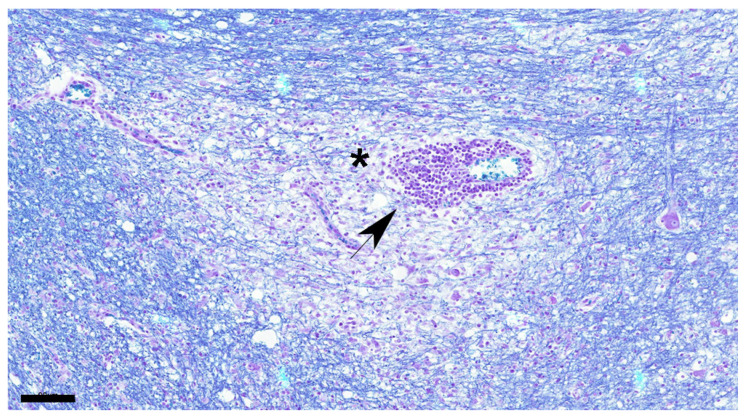
Demyelination is apparent around blood vessel (asterisk) showing cuffing (black arrow). Dog No. 11. Chronic group. LFB. Scale = 90 μm.

**Figure 6 viruses-16-01719-f006:**
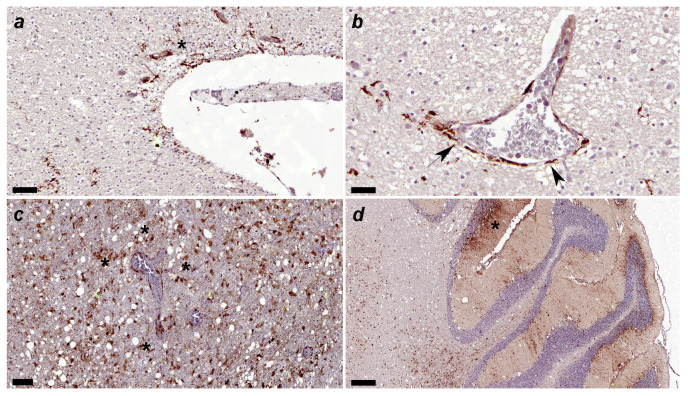
(**a**) Canine distemper virus (CDV)-immunopositive particles are detected in the subventricular parenchyma (asterisk). Immunohistochemical staining for CDV. No. 53. Acute group. Scale = 60 μm. (**b**) Canine distemper virus (CDV)-positive particles are observed mainly in astrocytic podocytes around blood vessel arrowhead. Vascular endothelial cells and intravascular mononuclear cells also show CDV-immunopositivity. Immunohistochemical staining for CDV. No. 53. Acute group. Scale = 30 μm. (**c**) Many canine distemper virus (CDV)-immunopositive particles are observed in glial cells around blood vessels. No. 249. Subacute 3 group. Scale = 100 μm. (**d**) Many canine distemper virus (CDV)-immunopositive particles are observed in the three layers (molecular, Purkinje cells, and granular) of gray matter and white matter (see asterisk in molecular layer). Immunohistochemical staining for CDV. No. 6. Subacute 3 group. Scale = 300 μm.

**Figure 7 viruses-16-01719-f007:**
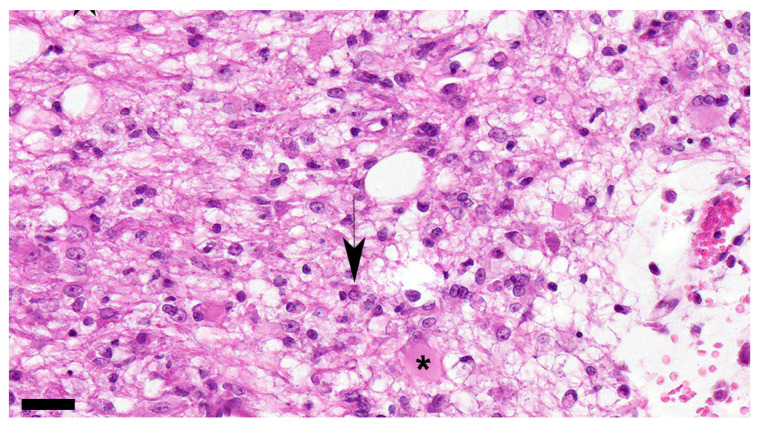
Marked increase in the number of astrocytes and microglia is detected in the white matter showing demyelination. Astrocytes are enlarged, recognized as gemistocytes (asterisk), and multinucleated cells are also seen. Nuclear inclusion bodies are observed (black arrow). Dog No. 436. Subacute 2 group. HE. Scale = 30 μm.

**Figure 8 viruses-16-01719-f008:**
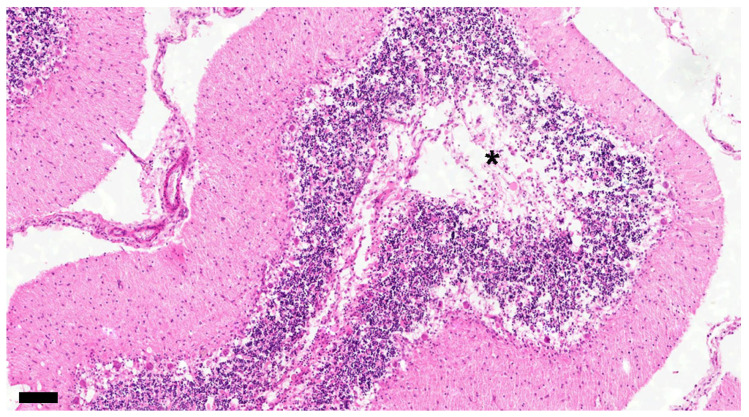
The white matter stains on some cerebellar folia have disappeared (asterisk). The number of cells in the granular layer decreases with decreasing number of Purkinje cells. No. 15. Chronic group. HE. Scale = 100 μm.

**Table 1 viruses-16-01719-t001:** Summary of primary antibodies and detection systems used.

Primary Antibody	Source	Dilution	Species/Type	Detection System
MCA 1893 Canine Distemper Virus	Biorad	1:250	Mouse/monoclonal	Mouse-on-canine HRP-Polymer
Glial Fibrillary Protein (GFAP)	Biocare Medical	RTU *	Mouse/monoclonal	MACH 4 Universal HRP-Polymer
Anti-Iba1 (ab5076)	Abcam	1:1000	Goat/polyclonal	Rabbit Anti-Goat IgG H&L (HRP) ab6741
Calbindin (orb5912)	Biorbyt	1:200	Rabbit/polyclonal	MACH 4 Universal HRP-Polymer

* Ready to use.

**Table 2 viruses-16-01719-t002:** Histological findings in cerebellum in each stage, including a summary of number of studied dogs and average age in each group.

Groups	Number of Animals	Average Month Age (Range)	Histopathological Findings
**Control**	5	(45 days old–6 years old)	No changes
**Acute**	4	(25 days old–3 years old)	glial cell reaction, inclusion bodies, CDV-immunoreactive particles, meningitis
**Subacute 1**	4	(30 days old–3 years old)	demyelination, glial cell reaction, inclusion bodies, CDV-immunoreactive particles, meningitis
**Subacute 2**	9	(2 months old–10 years old)	demyelination, glial cell reaction, inclusion bodies, CDV-immunoreactive particles, meningitis, mild decreased granular layer cells, and Purkinje cells
**Subacute 3**	5	(3 months old–3 years old)	demyelination, glial cell reaction, inclusion bodies, CDV-immunoreactive particles, meningitis, moderate decreased granular layer cells, and Purkinje cells
**Chronic**	6	(3 months old–3 years old)	demyelination, gliosis, glial cell reaction, inclusion bodies, CDV-immunoreactive particles, meningitis, severe decreased granular layer cells, and Purkinje cells

**Table 3 viruses-16-01719-t003:** Numbers of each cell type in cerebellum in each stage. Values (mean ± standard deviation) marked with different letters within the same column are significantly different (*p* < 0.05).

Groups	Number of Animals	Average Month Age (Range)	Numbers		
			Astrocytes	Microglia	Purkinje Cells
**Control**	5	(45 days old–6 years old)	35 ± 11 ^a^	64 ± 19	49 ± 6 ^a^
**Acute**	4	(25 days old–3 years old)	41 ± 12	64 ± 13	42 ± 17
**Subacute 1**	4	(30 days old–3 years old)	42 ± 13	56 ± 16	25 ± 6
**Subacute 2**	9	(2 months old–10 years old)	36 ± 13	75 ± 26	24 ± 14 ^b^
**Subacute 3**	5	(3 months old–3 years old)	32 ± 12	63 ± 26	15 ± 12 ^c^
**Chronic**	6	(3 months old–3 years old)	63 ± 22 ^b^	44 ± 17	23 ± 13 ^d^

## Data Availability

Data are contained within the article.

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
