# Peer review of "Pathological Study of Demyelination with Cellular Reactions in the Cerebellum of Dogs Infected with Canine Distemper Virus"

_viruses, 2024, doi:10.3390/v16111719_

Round 1

Reviewer 1 Report

Comments and Suggestions for Authors

The manuscript "Pathological study of demyelination with cellular reactions in the cerebellum of dogs infected with canine distemper virus" aims to examine the relationship between demyelination and glial/neuronal cell responses in the cerebellum of CDV-infected dogs. The manuscript is well structured and written. I think this article is very interesting and suitable for publication. However, I suggest some edits for improve the presentation of the manuscript:

Abstract

line 14 - change "glial cell responses" by "cellular reactions"

Introduction

line 35 and 51 - Please add this important reference ( SILVA, M.C. et al. 2009. Neuropathology of canine distemper: 70 cases (2005-2008). Pesq Vet Bras 29(8):643-652.

Material and methods

1st statment - Suggestion: "Twenty-eight dogs naturally infected by CDV were used in this study"

line 34 - change "cases" by "dogs"

lines 66-67 - "...infection in some dogs... PCR analysis." How many dogs ? Please specify

Lines 81-87 - I suggest to delete the parentheses after "subacute", here and in the all text. Use "subacute 1", "subacute 2", etc.

lines 93-94 - Why two monoclonal primary antibodies were used ? What the difference between immunohisotopathology and immunohistochemistry. I suggest to standardize these terms

line 106 - It would be interesting to say how many authors examined the samples to define the classification (blindly or no). 

Results

line 144 - "immunohistological" or immunohistochemical ?

line 146 - Title of the table is poor. Include more information here (details) such as animals ? organ ? groups ? etc.

Table 2 - What the histopathological difference between the groups subacute 2 and subacute 3? The findings are equal

Line 152 - Same comment of line 146. Are there signficant differences (p <0.05) among the values of the groups ? If yes, include a, b, c after the numbers

Line 197 - "choroid" or "choroid plexus" ? Clarify

Line 246 - In my opinion, this cell (asterisk) is a gemistocyte.  Do you agree ?

Line 259 - "cells". Glial cells ? Clarify

Line 261 - "granular layer". Granular layer is gray matter !!! In my opinion, molecular layer is very immunostained.

Discussion

Line 301 - "...this possibility". Which of them ? I think your results don't allow you to prove either of the two possibilities.

Line 331 - "...CDV infection." Chronic ? Do you agree ?

Author Response

Abstract:

Comment 1: line 14 - change "glial cell responses" by "cellular reactions"

Response 1: We agree with this comment. Done.

Introduction:

Comment 2: line 35 and 51 - Please add this important reference ( SILVA, M.C. et al. 2009. Neuropathology of canine distemper: 70 cases (2005-2008). Pesq Vet Bras 29(8):643-652.

Response 2: We agree with this comment, and include your recommended reference.

Material and methods:

Comment 3: 1st statment - Suggestion: "Twenty-eight dogs naturally infected by CDV were used in this study"

Response 3: We agree with this comment. Done.

Comment 4: line 34 - change "cases" by "dogs"

Response 4: We agree with this comment. Done.

Comment 5: lines 66-67 - "...infection in some dogs... PCR analysis." How many dogs ? Please specify

Response 5: We agree with this comments. These details were included according your recommendation.

Comment 6: Lines 81-87 - I suggest to delete the parentheses after "subacute", here and in the all text. Use "subacute 1", "subacute 2", etc.

Response 6: We agree with this comment. Done.

Comment 7: lines 93-94 - Why two monoclonal primary antibodies were used? What the difference between immunohisotopathology and immunohistochemistry. I suggest to standardize these terms

Response 7: Thank you for pointing this out. We have a mistake writing this, now was fixed. We standardize in all the manuscript as immunohistochemistry.

Comment 8: line 106 - It would be interesting to say how many authors examined the samples to define the classification (blindly or no).

Response 8: Thank you for pointing this out. We included the name of four veterinary pathologist that carried out the diagnosis. The diagnosis was carried out independently by each pathologist, working without knowing the diagnosis of the others three colleagues. Finally, they stablished a final consensus in each studied case.

Results

Comment 9: line 144 - "immunohistological" or immunohistochemical ?

Response 9: Thank you for pointing this out. We change to immunohistochemical in all the manuscript.

Comment 10: line 146 - Title of the table is poor. Include more information here (details) such as animals ? organ ? groups ? etc.

Response 10: We agree with this comment. Done.

Comment 11: Table 2 - What the histopathological difference between the groups subacute 2 and subacute 3? The findings are equal.

Response 11: Thank you for pointing this out. New details were included to better diferentiate the stages.

Comment 12: Line 152 - Same comment of line 146. Are there signficant differences (p <0.05) among the values of the groups ? If yes, include a, b, c after the numbers

Response 12: Thank you for pointing this out. Your recommendations were included.

Comment 13: Line 197 - "choroid" or "choroid plexus" ? Clarify

Response 13: Thank you for pointing this out. We have a mistake writing this, now was fixed as “choroid plexus”.

Comment 14: Line 246 - In my opinion, this cell (asterisk) is a gemistocyte.  Do you agree ?

Response 14: Yes, we agree!! Now, we include your recommendation in the figure caption, Thank you.

Comment 15: Line 259 - "cells". Glial cells ? Clarify

Response 15: Glial cells, was fixed. Thank you!

Comment 16: Line 261 - "granular layer". Granular layer is gray matter !!! In my opinion, molecular layer is very immunostained.

Response 16: Yes, We agree with this comment, and include your recommendations in figure captions.

Discussion

Comment 17: Line 301 - "...this possibility". Which of them ? I think your results don't allow you to prove either of the two possibilities.

Response 17: Thank you for pointing this out. We modified the sentence according your recommendation.

Comment 18: Line 331 - "...CDV infection." Chronic ? Do you agree ?

Response 18: Thank you for your comment. In our opinion, not only in chronic stage.

Reviewer 2 Report

Comments and Suggestions for Authors

This is a very good research paper on the evolution of canine distemper based on morphological and immunohistochemical studies. It is supported by exhaustive descriptions and excellent microphotographs.

The abstract is adequate and does not require any changes

The introduction presents updated information on the topic to be investigated. Although the article is part of a special issue on canine distemper, it would be interesting to include a general paragraph on the disease at the beginning of the introduction. The section, in general, is well written and introduces the topic. Only the sentence between lines 35-38 is confusing. It would be important to clarify what is called special staining. The authors later mention that they use luxol blue. Are they referring to this staining?

The materials and methods section describes the methodology very well. It would only be necessary to clarify whether the infection is natural (as it seems to be when reading the manuscript) or experimental. Rewrite the sentence between lines 71-72.

The description of results is exhaustive and includes photographs and graphs that illustrate the text very well. It would be necessary to include insets of the microphotographs of astrocytes in the figure corresponding to the samples in which GFAP was used

The discussion is very complete and analyzes the results in relation to the evolution of the disease. The authors demonstrate the importance of their descriptive results, rather than the qualitative ones, which is an example of the importance of deepening classical morphological studies of various diseases. In this case, it is observed how the progression of distemper is related to structural changes in the glial cells and not to their quantity.

As a minor aspect in Line 324, add species before animals

Author Response

REVIEWER 2: Responses to comments and suggestions for Authors

This is a very good research paper on the evolution of canine distemper based on morphological and immunohistochemical studies. It is supported by exhaustive descriptions and excellent microphotographs.

The abstract is adequate and does not require any changes

Comment 1: The introduction presents updated information on the topic to be investigated. Although the article is part of a special issue on canine distemper, it would be interesting to include a general paragraph on the disease at the beginning of the introduction. The section, in general, is well written and introduces the topic. Only the sentence between lines 35-38 is confusing. It would be important to clarify what is called special staining. The authors later mention that they use luxol blue. Are they referring to this staining?

Response 1: Thank you for your comment. The sentece was completely rewritten according to all your recommendations.

Comment 2: The materials and methods section describes the methodology very well. It would only be necessary to clarify whether the infection is natural (as it seems to be when reading the manuscript) or experimental. Rewrite the sentence between lines 71-72.

Response 1: Thank you for your comment. The sentece was rewritten according to your recommendation.

Comment 3: The description of results is exhaustive and includes photographs and graphs that illustrate the text very well. It would be necessary to include insets of the microphotographs of astrocytes in the figure corresponding to the samples in which GFAP was used.

Response 3: Thank you for your comment. These figures were redone according to your recommendation. Now we separated the Figure 1 in Figures 1, 2 and 3: In these three figures we include insets.

Comment 4: The discussion is very complete and analyzes the results in relation to the evolution of the disease. The authors demonstrate the importance of their descriptive results, rather than the qualitative ones, which is an example of the importance of deepening classical morphological studies of various diseases. In this case, it is observed how the progression of distemper is related to structural changes in the glial cells and not to their quantity.

Response 4: Thank you for your comment.

Comment 5: As a minor aspect in Line 324, add species before animals

Response 5: We agree with this comment. Done.

Reviewer 3 Report

Comments and Suggestions for Authors

Dear author,

Here are some suggestions and comments:

- In the Abstract there are several acronyms that are not identified. Acronyms should first be described in full before they can be used in the text.

- The citation is not correct in the text. References should be cited with numbering starting at 1 in the order in which they appear in the text. Please check the journal's guidelines and, if necessary, see other published articles for examples.

- In the Instroduction (Line 54) there are several acronyms that are not identified. Acronyms should first be described in full before they can be used in the text.

- The study objectives section is poorly written, with a very long and complicated to understand text.

- The Legend of Table 1 is poor.

- The ethical aspects of how the cadavers were provided were not mentioned. Furthermore, the approval protocol of the ethics committee for the use of animals is required.

- In the results, the authors limit themselves to referencing the tables throughout the text. The main results should be addressed more clearly and in text, leaving only minor data for the tables. By referring only to the tables, the authors force the readers to do all the work of extracting the main results. This needs to be reviewed.

- Figure 1 is poorly constructed. Separate the graphs from the histopathological images. Increase the resolution of the microscopy and indicate in the figures what should be highlighted. The caption for Figure 1 is horrible, as much of the text that should be highlighted in the results was placed in caption format.

- Figure 2 and 6 does not have the necessary markings.

- The scale bars of the figures cannot be clearly seen.

- The article has excellent results, which however need better presentation.

Author Response

REVIEWER 3: Responses to comments and suggestions for Authors

Comment 1: In the Abstract there are several acronyms that are not identified. Acronyms should first be described in full before they can be used in the text.

Response 1: We agree with this comment. Done.

Comment 2: The citation is not correct in the text. References should be cited with numbering starting at 1 in the order in which they appear in the text. Please check the journal's guidelines and, if necessary, see other published articles for examples.

Response 2: We agree with this comment. Fixed.

Comment 3: In the Instroduction (Line 54) there are several acronyms that are not identified. Acronyms should first be described in full before they can be used in the text.

Response 3: We agree with this comment. Fixed.

Comment 4: The study objectives section is poorly written, with a very long and complicated to understand text.

Response 4: We agree with this comment. Fixed.

Comment 5: The Legend of Table 1 is poor.

Response 5: We agree with this comment. We improve the legend of Figure 1. Please, confirme if now is good enough. Thank you.

Comment 6: The ethical aspects of how the cadavers were provided were not mentioned. Furthermore, the approval protocol of the ethics committee for the use of animals is required.

Response 6: Thank you for pointing this out. We included this information. Please, confirm if now is according to your recommendation.

Comment 7: In the results, the authors limit themselves to referencing the tables throughout the text. The main results should be addressed more clearly and in text, leaving only minor data for the tables. By referring only to the tables, the authors force the readers to do all the work of extracting the main results. This needs to be reviewed.

Response 7: Thank you for pointing this out. We have done an effort to consider your recommendations, but in part, we prefer to preserve the specific information of the stage groups in the original table. Please, confirm if you agree with this new version.

Comment 8: Figure 1 is poorly constructed. Separate the graphs from the histopathological images. Increase the resolution of the microscopy and indicate in the figures what should be highlighted. The caption for Figure 1 is horrible, as much of the text that should be highlighted in the results was placed in caption format.

Response 8: We greatly appreciate your comment. These figures were completely redrawn according to your recommendation. Now we separated the Figure 1 in Figures 1, 2 and 3: Furthermore, in these three figures we include insets. Thank you.

Comment 9: Figure 2 and 6 does not have the necessary markings.

Response 9: We agree with this comment. Fixed.

Comment 10: The scale bars of the figures cannot be clearly seen.

Response 10: We agree with this comment. Fixed.

Round 2

Reviewer 3 Report

Comments and Suggestions for Authors

I am pleased with the corrections and the opportunity to help the authors bring a better version of the manuscript.